validation; suicide prevention; questionnaire; specialists; postvention

**Corresponding author:**
Lai Fong Chan;
Email: laifchan@ppukm.ukm.edu.my

# The Advanced C.A.R.E. Suicide Prevention Gatekeeper Training Questionnaire for medical lecturers and specialists: A psychometric evaluation in a Malaysian sample

Iman Mohamed Ali[1] [iD], Lai Fong Chan[2], Tuti Iryani Mohd Daud[2], Abdul Razak Othman[3], Yin Ping Ng[4], Kai Shuen Pheh[5] and Ravivarma Rao Panirselvam[6]

[1]Department of Psychiatry, Sarawak General Hospital, Kuching, Malaysia; [2]Department of Psychiatry, Faculty of Medicine, National University of Malaysia, Kuala Lumpur, Malaysia; [3]Department of Psychiatry, Kuala Lumpur General Hospital, Kuala Lumpur, Malaysia; [4]Psychiatry Specialty, Pantai Hospital Penang, Bayan Baru, Malaysia; [5]Department of Psychology and Counselling, Faculty of Arts and Social Sciences, Tunku Abdul Rahman University, Kampar, Malaysia and [6]Department of Psychiatry, Miri Hospital, Miri, Malaysia

## Abstract

Given the high prevalence rate of suicidal ideation amongst medical students, medical lecturers and specialists as gatekeepers should be well-trained in suicide prevention. There is a need for validated measures to assess gatekeeper training gains for suicide prevention. The psychometric properties of the Advanced C.A.R.E. Suicide Prevention Gatekeeper Training Questionnaire (AdCARE-Q) were evaluated for a sample of medical lecturers and specialists in Malaysia. A total of 120 participants completed 24 items in the AdCARE-Q. Analysis of variance of perceived knowledge scores was performed. Exploratory factor analysis (EFA) was conducted. Reliability was calculated. The AdCARE-Q was reduced to 15 items that fit into two factors, "self-efficacy" and "declarative knowledge." Overall internal consistency was good with Cronbach's alpha = 0.84. The intraclass correlation coefficient between groups from the psychiatry department and non-psychiatry departments was good at 0.80. The oldest age group and participants from the Psychiatry department scored significantly higher than other groups in perceived knowledge of suicide prevention. This study found that the AdCARE-Q has adequate psychometric properties to assess suicide prevention gatekeeper training gains amongst medical lecturers and specialists. Confirmatory factor analysis is recommended for future studies.

## Impact statement

Many healthcare professionals work to prevent suicide amongst their patients and the community. However, in the effort to prevent suicide, healthcare professionals frequently overlook the risks of suicide within the medical fraternity itself. The medical field – whether in the study or practice of medicine – has been regarded by many as extremely challenging and a significant source of stress. The nature of health care itself, with the multiple stressors faced in the service of ill patients, who can be demanding and unempathetic, is a major source of stress for healthcare professionals. High rates of depression and suicidal ideation can be found in medical students. Suicide prevention strategies to assist healthcare professionals should focus on suicide prevention amongst medical students as they would form the future medical fraternity. The Advanced C.A.R.E. Suicide Prevention Gatekeeper Training program was developed for gatekeepers that include medical lecturers and specialists. This study aims to study the psychometric properties of the AdCARE-Q, that it may be used to assess the efficacy of and knowledge gains from the training program. The questionnaire includes assessment of knowledge on safety planning and suicide postvention as well, which had not been included in questionnaires assessing suicide prevention training prior to this study being conducted.





## Introduction

Despite being advocates of suicide prevention, depression and suicidal risk does not escape the medical fraternity (Mata et al., 2015, 2016; Rotenstein et al., 2016). Exposure to the nature of health care, the multiple stressors faced in the service of patients, the frequent necessity to keep updated in order to remain relevant and the lack of acknowledgment may all contribute to risks of mental health issues amongst medical professionals which could lead to suicide (Mata et al., 2015,

2016; Rotenstein et al., 2016). Suicide prevention strategies to assist healthcare professionals should focus on suicide prevention amongst medical students as they would form the future medical faculty. A study (Rotenstein et al., 2016) found that the prevalence of suicidal ideation amongst medical students is 11.1%. The study also found that the percentage of medical students who screened positive for depression that sought psychiatric treatment was only 15.7%. This is concerning as it shows an increased risk for suicide amongst medical trainees who do not seek treatment.

Gatekeepers in the medical fraternity would include medical lecturers and specialists, as they educate and work with medical students, interns and junior doctors. Suicide prevention gatekeeper training programs are developed to train gatekeepers in recognizing and eliciting the warning signs and risks for suicide, methods on approaching an individual with suicide intention, persuading persons-at-risk to seek assistance and making referrals to the appropriate resources (Davis et al., 2006; Tompkins et al., 2010; Bean and Baber, 2011; Arensman et al., 2016; Teo et al., 2016; Terpstra et al., 2018). Gatekeeper training has been employed in various settings, ranging from schools to the militaries and healthcare professionals, showing significant improvement in domains of knowledge, self-efficacy and confidence to act (Tompkins et al., 2010; Bean and Baber, 2011; Arensman et al., 2016; Teo et al., 2016) passing on the training information, and increased the referrals of youths at-risk to the appropriate services (Matthieu et al., 2008; Wyman et al., 2008; Cross et al., 2011; Rodi et al., 2012; Susanne Condron et al., 2015). However, these studies did not measure attitudes of the participants toward suicide or the willingness of participants to act with a suicidal person, targeting only cognitive gains (Osteen et al., 2014).

A systematic review (Davis et al., 2006) found that health professionals tend to over-estimate their self-assessments of competence. A study (Siau et al., 2018) found that gains in declarative knowledge were not maintained by healthcare professionals at 3-month follow-up, although gains in perceived knowledge were maintained. Thus, a questionnaire that assesses not only perceived but also declarative or tested knowledge would be more reliable in assessing outcomes of gatekeeper training programs.

The C.A.R.E. program is a locally developed suicide prevention program (Pheh et al., 2019). The training program educates laypersons on catching the warning signs of suicidal behaviors early (C), acknowledging emotions (A), building empathy and appropriately responding to people with suicidal behavior, reviewing risk factors (R) and encouraging the person at risk to receive professional assistance (E). The Advanced C.A.R.E. suicide prevention gatekeeper training is an enhanced version of the basic C.A.R.E. program targeted at healthcare professionals with emphasis on prevention rather than prediction; by teaching the recognition of suicide risks and approaches to reduce such risks. The Advanced C.A.R.E. also incorporated the six-step safety planning intervention (Stanley et al., 2018) and suicide postvention in its program. Suicide postvention includes strategies to prevent suicide amongst people who have been bereaved by suicide, that is, individuals or communities that have experienced the loss of someone by suicide who may be at an increased risk of suicide themselves (Andriessen et al., 2019).

The Advanced C.A.R.E. Suicide Prevention Gatekeeper Training Questionnaire (AdCARE-Q) was developed to assess the efficacy of the Advanced C.A.R.E. program. The AdCARE-Q attempts to measure gains in the domains of knowledge on suicide risk factors, signs and behavior, attitudes toward suicide and persons at-risk and the confidence and ability to practice the knowledge gained and provide appropriate referrals of persons at risk. In addition, AdCARE-Q included items assessing participants about safety planning and suicide postvention measures.

The development of the AdCARE-Q may assist in the revision and expansion of content of suicide prevention gatekeeper training programs where necessary. This study aims to evaluate the psychometric properties of the AdCARE-Q in a Malaysian sample of medical lecturers and specialists as gatekeepers in the medical fraternity.

## Methods

This validation study was based on construct validity and conducted exploratory factor analysis (EFA). The study was conducted in three settings – the medical faculty of a public university, a public tertiary teaching hospital and a tertiary general hospital – all located in Kuala Lumpur, the capitol city of Malaysia.

For EFA, the rule of thumb suggested ratio is at least five subjects to each item (Gorsuch, 1983). The AdCARE-Q consists of 24 items. Thus, a total of 120 participants were recruited for this study from October to December 2021. Participants were medical faculty lecturers and medical specialists, regardless of field or specialty. Only lecturers and specialists who had performed any amount of clinical supervision of undergraduate or postgraduate medical students and those who were sufficiently literate in the English language, to facilitate understanding of the AdCARE-Q, were selected. All subjects were provided with an information sheet and assured of the confidentiality of their personal details and data submitted. All participants provided their informed consent prior to study entry. Those who were not medical lecturers or specialists, who had no experience training medical students, and had not given their informed consent were not included for this study. Participants were recruited via purposive sampling from the three settings.

Reliability of the AdCARE-Q was also measured in this sample using Cronbach's alpha, and intraclass correlation coefficient was calculated to neutralize the findings from the Psychiatry department respondents.

### Study instruments

#### Sociodemographic questionnaire
Demographic data were collected from the participants, which included the subjects' age range, race, profession and medical field or specialty.

#### AdCARE-Q
The AdCARE-Q was adapted from the questionnaire used to assess knowledge gains in a suicide prevention gatekeeper training study (Terpstra et al., 2018). Written permission to adapt this questionnaire for this research was obtained from the author, Dr. Renske Gilissen of Leiden University, Holland. The questionnaire used in the study (Terpstra et al., 2018) had not been validated.

AdCARE-Q is a self-administered questionnaire which uses the five-point Likert-type scale with a response scale from 1 to 5 for each question to reflect the level of confidence of the subject in responding to the domains of knowledge (K), attitudes (A) and practice (P) in suicide prevention. The AdCARE-Q utilizes the English language medium.

The knowledge domain assesses recognition and identification of warning signs of suicidal behavior and risk factors. The attitudes

domain assesses the subject's beliefs toward suicide and empathy toward people at-risk. The practice domain assesses the confidence, willingness and ability of the subject to assist people at-risk and refer them to the appropriate channels. An item assessing suicide postvention knowledge was added to the AdCARE-Q. Earlier studies involving evaluations of suicide prevention training programs (Davis et al., 2006; Tompkins et al., 2010; Bean and Baber, 2011; Arensman et al., 2016; Teo et al., 2016; Terpstra et al., 2018) had not included items about suicide postvention.

### Procedure

#### Development of the AdCARE-Q

The questionnaire (Terpstra et al., 2018) consisted of 10 items – three items measuring the practice of suicide prevention measures, four items measuring perceived (self-rated) knowledge of suicide prevention, and confidence in implementing suicide prevention measures. This questionnaire was adapted into the AdCARE-Q by adding, removing and altering items, in keeping with the three domains measured by the AdCARE-Q.

An expert consensus meeting was performed amongst the research authors who were experienced in the development, implementation and evaluation of local gatekeeper training programs.

The AdCARE-Q attempted to assess three domains – knowledge (K), attitudes (A) and practice (P) of suicide prevention. The knowledge domain is assessed with 11 items B1, B2, C1, C2, C3, C4, C5, C6, C9, C14 and D1. Item C9 assesses knowledge of safety planning measures.

The attitudes domain is assessed with six items B3, C7, C8, C10, C15 and D2. Items C7 and C15 were added to assess beliefs toward suicide prevention, while items C8 and C10 were added to assess ability to acknowledge emotions and empathize.

The practice domain is assessed with seven items B4, C11, C12, C13, D3, D4 and D5. Item D4 assesses confidence in discussing safety planning with persons at-risk. Item D5 assesses suicide postvention knowledge and the related referral.

In the AdCARE-Q, Sections B and D assess perceived knowledge and abilities pertaining to suicide prevention. Section C was added to assess declarative (tested) knowledge on suicide prevention. This is to ensure a more accurate and objective estimate of knowledge gains from the suicide prevention training program (Supplementary Material).

Thus, a total of 24 items were included in the AdCARE-Q, prior to factor analysis (Table 1).

#### Face validation

In June 2021, face validation was conducted with a sample of 10 subjects consisting of nine medical officers (five from Anesthesiology, one from General Surgery, one from Ophthalmology, one from Radiology and one from Public Health departments) and one medical assistant (from Anesthesiology department) from a teaching hospital who were not from the Psychiatry department. This was to ensure that even participants who were not well-versed in psychiatry and were less specialized than medical lecturers or specialists were able to understand the AdCARE-Q. They were informed about the objectives of this study and were required to complete the digitized form which included the information sheets, consent forms, socio-demographic questionnaire and the AdCARE-Q. They were required to give feedback on their ability to understand the language and content of the AdCARE-Q and respond to all the items appropriately. All 10 responses indicated that the AdCARE-Q was deemed acceptable and no item revisions

were needed. Each participant spent 10–15 min to complete the questionnaires. This sample that was used for the face validation was not included in the subsequent stages of this study.

### Data collection

Due to the COVID-19 pandemic, medical faculty lecturers and specialist doctors from the public university, teaching hospital and general hospital were sampled using an online form to conduct this validation study. Due to the COVID-19 pandemic, sampling was communicated via email, organizational or institutional email list server and mailing lists, instant messaging applications or face-to-face meetings. The information sheets about this study, consent form for participation in this study, socio-demographic data form and the AdCARE-Q were all included in the digitized form. All data had been anonymized and kept confidential.

### Results

#### Descriptive statistics

A total of 120 participants were included in this study. All participants met the inclusion criteria, and none were excluded (Table 2).

The majority of the participants were from the general hospital with a total of 68 subjects (56.7%), followed by the teaching hospital with 44 subjects (36.7%), and the public university with 8 subjects (6.7%). There was an almost equal representation of the sexes with 68 female participants (56.7%), and 52 male participants (43.3%). The majority of the participants were Malay (70 participants, 58.3%), followed by Chinese (35 participants, 29.2%) and Indian (15 participants, 12.5%). Most of the participants were between 41 and 50 years of age (60%), followed by 35.8% of participants who were 31 and 40 years of age.

With regards to profession, 77.5% (93 participants) of the sample were medical specialists, 17.5% (21 participants) worked as both medical specialists and lecturers, and 5% (6 participants) were medical lecturers. The highest number of respondents were from the Psychiatry department (24 participants, 20%), followed by the General Medicine department (20 participants, 16.7%), and the Anesthesiology and Surgery departments (10 participants, 8.3%). The least number of respondents were from the Biochemistry, Urology, Infectious Diseases, Nursing, Respiratory Medicine, Rheumatology and Oncology departments, recording only one respondent per department. This was followed by the Radiology, Neurology, Medical Microbiology and Immunology, Family Medicine and Anatomy departments with two respondents per department.

Comparison between gender and perceived knowledge scores for suicide prevention (item B1), showed no significant differences between male and female groups ($p > 0.05$). The mean scores for both the gender groups were also closely similar, with the male group mean scores of 2.81 and the female group mean scores of 2.74.

When comparing between the age groups of the participants and scores on perceived knowledge about suicide prevention, significant difference was found between the age groups ($p < 0.01$; Table 3). The older age group of 51–60 years old scored the highest for perceived knowledge (mean scores = 3.80), whereas the other age groups of 31–40 years old (mean scores = 3.02) and 41–50 years old (mean scores = 2.54) scored lower.

**Table 1.** Adaptation of the AdCARE-Q

| Factor: Knowledge | New items | Terpstra study questionnaire items |
|---|:---:|:---:|
| B1 Knowledge on suicide prevention | √ | |
| B2 Knowledge on warning signs of suicide | | √ |
| C1 Most people who attempt suicide show warning signs before their attempt. | √ | |
| C2 Suicide is usually caused by more than one factor. | √ | |
| C3 Only people who have been diagnosed with mental illness are at risk of suicide. | √ | |
| C4 Depression is a potential suicide risk. | √ | |
| C5 Making final plans or giving away prized possessions are warning signs for suicide. | √ | |
| C6 People who talk about committing suicide are less likely to attempt suicide. | √ | |
| C9 Understanding the method of suicide in a person who is suicidal is necessary for safety planning. | √ | |
| C14 Describing explicit details about suicidal methods in the media is harmful. | √ | |
| D1 I have confidence in my abilities to recognize warning signs of suicide in people. | √ | |
| **Factor: Attitudes** | | |
| B3 Communicating with someone who is suicidal | | √ |
| C7 Asking about suicidal thoughts will cause a person to develop suicidal ideas. | √ | |
| C8 People who are suicidal may not see a way out of their problems. | √ | |
| C10 Acknowledging a suicidal person's distress should be done before offering any advice. | √ | |
| C15 I must consider my own safety when attending to a suicidal person. | √ | |
| D2 I hesitate to ask a person whether they are suicidal | | √ |
| **Factor: Practice** | | |
| B4 How to arrange help for a suicidal person | | √ |
| C11 A person who shows warning signs of suicide should be referred to a healthcare provider. | √ | |
| C12 We must never disclose a person's suicidal plan without their permission. | √ | |
| C13 Crisis helplines should be offered to a suicidal person. | √ | |
| D3 I have confidence in my abilities to arrange for help for someone who is suicidal. | √ | |
| D4 I am confident in discussing about safety planning with someone who is suicidal. | √ | |
| D5 I know where to seek resources for postvention services. | √ | |

*Note*: New items and items from the questionnaire used in the Terpstra study (2018) grouped according to the proposed three factors: knowledge, attitudes and practice.

Participants who were both medical specialists and lecturers scored slightly higher in perceived knowledge (mean scores = 3.10) about suicide prevention than medical specialists (mean scores = 2.68) or medical lecturers (mean scores = 3.00). However, when using analysis of variance (ANOVA), these differences were not significant ($p > 0.05$). Similarly, there was no significant difference between scores for perceived knowledge ($p > 0.05$) when compared to the participants' workplaces of Public University (mean scores = 3.13), Teaching Hospital (mean scores = 2.80) and General Hospital (mean scores = 2.71).

Comparisons between the specialties of participants when scoring perceived knowledge showed significant differences ($p < 0.01$; Table 4). Participants from the departments with the highest number of respondents were compared. Participants from the Psychiatry department showed significantly higher mean scores of 4.00 for perceived knowledge about suicide prevention compared to participants from the General Medicine (mean scores = 2.35), Anesthesiology (mean scores = 2.40) and Surgery (mean scores = 2.40) departments.

### Exploratory factor analysis (EFA)

On extraction with principal component analysis (PCA), the items that had low communalities (<0.5) were removed as they had considerable variance unexplained by the extracted factors. Eigen values of the remaining items were calculated and the number of factors were determined based on the total variance explained by these factors correlating with eigen values of >1. On varimax rotation with Kaiser normalization, more items that had overlapping factor loading were removed. On repeated PCA extractions, a total of 10 items were excluded. The AdCARE-Q was reduced to 14 items.

All the remaining 14 items (B1–B4, C4, C8, C10, C11, C13, D1–D5) had communalities values of >0.5 (Table 5). Eigen values for these 14 items ranged from 0.052 to 7.057 and two factors explained 69.8% of the total variance (Table 5). Scree plot showed the drop in eigen values after the second factor. Thus, two factors were extracted for the AdCARE-Q. On varimax rotation with Kaiser normalization with these 14 items, all the 14 items showed good

**Table 2.** Characteristics of 120 participants in the study

|  | Total (N) | Percentage (%) |
|---|---|---|
| Sex |  |  |
| Male | 52 | 43.3 |
| Female | 68 | 56.7 |
| Age |  |  |
| 31–40 years old | 43 | 35.8 |
| 41–50 years old | 72 | 60.0 |
| 51–60 years old | 5 | 4.2 |
| Race |  |  |
| Malay | 70 | 58.3 |
| Chinese | 35 | 29.2 |
| Indian | 15 | 12.5 |
| Profession |  |  |
| Medical specialist | 93 | 77.5 |
| Medical faculty lecturer | 6 | 5.0 |
| Medical specialist/medical lecturer | 21 | 17.5 |
| Organization |  |  |
| Public University | 8 | 6.7 |
| Teaching Hospital | 44 | 36.7 |
| General Hospital | 68 | 56.7 |
| Department |  |  |
| Psychiatry | 24 | 20.0 |
| General Medicine | 20 | 16.7 |
| Anesthesiology | 10 | 8.3 |
| Surgery | 10 | 8.3 |
| *Others* | 56 | 46.7 |

**Table 3.** Association between respondents' age groups and perceived knowledge on suicide prevention (item B1 scores)

| Age groups | Mean | Standard deviation | F | Mean square | p |
|---|---|---|---|---|---|
| 31–40 years | 3.02 | 1.144 | 6.420 | 5.907 | **0.002*** |
| 41–50 years | 2.54 | 0.838 |  |  |  |
| 51–60 years | 3.80 | 0.837 |  |  |  |

*$p < 0.05$ for ANOVA.

factor loadings onto each of the two extracted factors of >0.5 (Table 5). Nine items loaded on to factor 1 (B1–B4, D1–D5), and five items on factor 2 (C4, C8, C10, C11, C13).

Kaiser–Meyer–Olkin (KMO) measure of sampling adequacy value was found to be good at 0.892 (>0.6) and Bartlett's test of sphericity was significant at $p$-value <0.01.

## Reliability

Cronbach's alpha was calculated for the 14 items and showed good internal consistency at 0.84, indicating that these items measured the same constructs. There was good internal consistency for both factor 1 with nine items (Cronbach's alpha = 0.852) and factor 2 with five items (Cronbach's alpha = 0.825). The intraclass correlation coefficient comparing responses from psychiatrists and non-psychiatrists showed a good reliability (Koo and Li, 2016) of 0.797 ($p < 0.01$), inferring that the AdCARE-Q can be considered a reliable tool for both psychiatrists and non-psychiatrists.

## Discussion

On initial development of the AdCARE-Q, there were a proposed number of three domains or factors to be assessed – knowledge (K), attitudes (A) and practice (P) of suicide prevention. However, after EFA, only two factors remained with a total of 14 items.

Items such as C5 "Making final plans or giving away prized possessions are warning signs for suicide" and C6 "People who talk about suicide are less likely to attempt suicide" were surprisingly removed, as they showed low impact on assessing the extracted factors despite being important warning signs for suicide. This could reflect the sample pool that consisted mostly of medical professionals that did not have a psychiatric background and less experience with suicide prevention, as these items were relatively more complex compared to the items that remained (Siau et al., 2017). The wording and complexity of these items would result in a wider variance of responses, affecting their relationship with the extracted factors. With regards to item C5, people may make provisions for their families or manage their assets and end of life care, without the presence of suicidal ideation. A study (Tilse et al., 2016) found that 59% of the general adult population had made wills. In a Malaysian population with its many cultural differences (Siau et al., 2017), certain cultural norms encourage bequeathing or the division of assets (Salisu, 2017). Thus, for item C5, "making final plans or giving away prized possessions" may not be construed by some respondents as a warning sign for suicide. Perhaps a more culturally appropriate representation of this construct could include the terms "suddenly sending final farewell messages" or "asking for forgiveness." For item C6, pre-conceived beliefs and attitudes toward persons who are suicidal may have affected the responses as well (Renberg and Jacobsson, 2003; Hjelmeland et al., 2006; Siau et al., 2017; Zahiruddin et al., 2018; Goni et al., 2020). It is a common, albeit false, belief that people who express suicidal intentions are only seeking attention and are not serious about attempting suicide and a study (Saini et al., 2016) found that these beliefs were held by doctors interviewed about managing suicidal patients. Another study (Pisani et al., 2011) found attitudes toward suicide prevention difficult to change due to its multifactorial nature.

One of the items assessing knowledge on safety planning measures, C9 "Understanding a suicidal person's method of suicide is necessary for safety planning" was also removed. Safety planning concepts in suicide prevention may appear to be a foreign subject amongst respondents with no psychiatric background and lesser exposure to suicide prevention. This could also explain the reason behind the discrepancy where another item D4, assessing the self-perceived confidence of the respondent about "discussing safety planning" was finally retained in the AdCARE-Q. Item D4 simply presented safety planning as a general topic with a self-rated ability, without testing the knowledge on more specified concepts included in safety planning.

**Table 4.** Association between respondents' department/specialty and perceived knowledge on suicide prevention (item B1 scores)

| Department | Mean | Standard Deviation | F | *t*-value | *p* |
|---|---|---|---|---|---|
| Psychiatry | 4.00 | 1.049 | 30.030 | 12.628 | **0.000*** |
| Non-psychiatry | 2.46 | 0.971 | | | |

| Specialty/Department | Mean | Standard Deviation | F | Mean Square | *p* |
|---|---|---|---|---|---|
| Psychiatry | 4.00 | 0.417 | 18.087 | 11.534 | **.000**** |
| General Medicine | 2.35 | 0.813 | | | |
| Anaesthesiology | 2.40 | 0.843 | | | |
| Surgery | 2.4 | 0.843 | | | |
| Others | 2.52 | 0.894 | | | |

*\*p*-value <0.05 for *t*-test.
*\*\*p*-value <0.05 for ANOVA.

**Table 5.** Exploratory factor analysis

| Items | Explained variance (%) | Factor loading | Communality |
|---|---|---|---|
| **Total** <br> **Factor 1** | **69.791** <br> **50.408** | | |
| D1 I have confidence in my abilities to recognize warning signs of suicide in people. | | 0.940 | 0.890 |
| B2 Knowledge on warning signs of suicide | | 0.937 | 0.881 |
| B1 Knowledge on suicide prevention | | 0.903 | 0.822 |
| D4 I am confident in discussing about safety planning with someone who is suicidal. | | 0.890 | 0.798 |
| B3 Communicating with someone who is suicidal | | 0.889 | 0.792 |
| D3 I have confidence in my abilities to arrange for help for someone who is suicidal. | | 0.810 | 0.686 |
| D2 I hesitate to ask a person whether they are suicidal. | | −0.797 | 0.641 |
| D5 I know where to seek resources for suicide postvention services. | | 0.768 | 0.640 |
| B4 How to arrange help for someone who is suicidal | | 0.766 | 0.602 |
| **Factor 2** | **69.791** | | |
| C4 Depression is a potential suicide risk. | | 0.824 | 0.686 |
| C8 People who are suicidal may not see a way out of their problems. | | 0.800 | 0.640 |
| C13 Crisis helplines should be offered to a person who is suicidal. | | 0.744 | 0.599 |
| C11 A person who shows warning signs of suicide should be referred to a healthcare provider. | | 0.726 | 0.528 |
| C10 Acknowledging the distress of a person who is suicidal should be done before offering any advice. | | 0.716 | 0.564 |

*Note*: Explained variance, factor loadings, and communalities based on a principal components analysis with varimax rotation for 14-item AdCARE-Q (*N* = 120).

After EFA, items B1 to B4 and D1 to D5, were all included in factor 1. These nine items measure perceived knowledge on suicide prevention and confidence in practicing suicide prevention. Items B3, B4 and D1 to D4 all attempt to evaluate self-rated abilities and confidence about suicide prevention. However, items B1, B2 and D5 evaluate self-rated knowledge. Thus, not all the items in factor 1 fall under the definition of the practice or knowledge factors. Therefore, factor 1 was re-named as "self-efficacy" to better represent these items as a whole. The self-efficacy factor is defined as perceived knowledge about suicide prevention and the confidence about the ability and willingness to execute suicide prevention measures. Similarly, other studies

(Wyman et al., 2008; Siau et al., 2018) also used similar items in their questionnaire which attempted to measure self-rated efficacy of gatekeepers.

After EFA, the items C4, C8, C10, C11 and C13 were included in factor 2. Only item C4 "Depression is a potential suicide risk" had been proposed to be in the knowledge factor prior to factor analyses. Items C8 "People who are suicidal may not see a way out of their problems" and C10 "Acknowledging the distress of a person who is suicidal should be done before offering any advice" were initially included in the attitudes factor. Items C11 "A person who shows warning signs of suicide should be referred to a healthcare provider" and C13 "Crisis helplines should be offered to a person

who is suicidal" were initially proposed to be in the practice factor. The five items that remained in factor 2 were based on factual knowledge of suicide prevention and not self-rated ability, similar to other studies (Wyman et al., 2008; Siau et al., 2018). These items also did not clearly fit into either the practice or the attitudes factor as a whole unit. Thus, factor 2 was renamed as "declarative knowledge" to differentiate it from the self-rated efficacy and perceived knowledge items in factor 1. The declarative knowledge factor was redefined as tested knowledge about warning signs and risk factors for suicide and referrals to the appropriate channels. This is a necessary factor in suicide prevention training questionnaires as self-rated knowledge may not necessarily concur with actual tested knowledge scores (Wyman et al., 2008).

Therefore, after EFA, this study found that the 14 items that remained in the AdCARE-Q were better explained by the two factors of "self-efficacy" and "declarative knowledge." A total of 10 items, all from the tested or declarative knowledge category, were removed. Alternately, all the self-rated efficacy items were included. This would indicate a discrepancy between the self-rated and tested knowledge (Wyman et al., 2008). The five remaining declarative knowledge items test most of the concepts in the Advanced C.A.R.E. program, such as recognizing risk factors, acknowledging emotions, catching the warning signs and encouraging adequate referrals. The self-efficacy items on suicide postvention and safety planning were retained after factor analysis. These concepts may be relatively new and unfamiliar to gatekeepers who may have not have received adequate training (Moscardini et al., 2020), which could have led to the declarative knowledge item on safety planning being removed. The aim of the training program would be to expose gatekeepers to these insights and knowledge that may yet be unfamiliar to them. The five remaining declarative knowledge items also test certain steps in safety planning which include recognizing warning signs and linking with the appropriate assistance (Stanley et al., 2018). Thus, the questionnaire would still be adequate to test the knowledge gains during the training program.

On reliability analysis, the adapted questionnaire with 14 items showed a good overall internal consistency. Furthermore, there was good internal consistency within the two individual factors. The intraclass correlation coefficient findings show that the AdCARE-Q has good reliability when used by both psychiatrists and non-psychiatrists groups.

With regards to the descriptive statistical findings in this study, the older age group of 51–60 years old scored significantly higher for perceived knowledge, whereas the other age groups of 21–30 years old and 31–40 years old scored lower. There was not much difference between the average scores of the two younger age groups. In the field of medicine, age generally signifies number of years of service or experience. The larger amount of experience with suicidal cases due to longer time spent in the field of medicine, would make medical professionals in the older age group more confident about their knowledge on suicide prevention. A study on the association of surgeon age and experience with heart surgery outcomes found that age was highly correlated with years of experience since graduation and since fellowship (Anderson et al., 2017). Another study found that senior physicians self-reported stronger competencies in health informatics than their more junior counterparts, although junior physicians scored higher in the usage of health informatics (Devitt and Murphy, 2004). Similarly, more experience handling suicidal cases is reflected when the participants from the Psychiatry department showed significantly higher average scores for perceived knowledge about suicide prevention compared to participants from the other departments, similar to the findings in another study (Siau et al., 2018). This would also suggest future studies excluding participants from the Psychiatry department to ensure a better result reflecting knowledge amongst untrained gatekeepers in the medical field.

There were no significant differences of average scores on perceived knowledge across professions of specialist doctors and medical lecturers, and their workplaces. As this study involves all urban settings with established centers, and professionals with an expected higher amount of medical knowledge (specialists and lecturers), there would be no significant difference in these aspects.

## Strengths and limitations

This study achieved heterogeneity of participants according to age range, profession and departments, in accordance with the study objectives of targeting gatekeepers in the medical faculty. The AdCARE-Q also had good internal consistencies. To the best of our knowledge, the AdCARE-Q is the first published suicide prevention gatekeeper training questionnaire to include awareness of suicide postvention, which is an important aspect of suicide prevention.

This study was conducted via purposive sampling to recruit participants. The study was also confined to only three centers, and all were urban in setting, therefore the findings are not fully representative of medical specialists and lecturers nationally in Malaysia. A majority of the respondents did not come from a psychiatric background which could be projected as not having much experience managing suicide prevention, and thus could reflect their varied understanding of the more complex items in the questionnaire. This study also did not exclude participants who had already received training for suicide prevention. These participants could either create a bigger discrepancy with the removed items that were unfamiliar to the untrained participants, or could affect the validity of the items that were finally included in this questionnaire. Future studies excluding participants from the Psychiatry department and participants who have received suicide prevention training are also warranted to better reflect results from gatekeepers untrained in suicide prevention. Confirmatory factor analysis in a different sample is recommended for future studies to improve the psychometric properties of the AdCARE-Q. Test–retest reliability was also not conducted for this study.

## Conclusion

This study demonstrated that the AdCARE-Q has adequate psychometric qualities as a measure of suicide prevention training gains for gatekeepers, namely medical lecturers and specialists in Malaysia. With the use of the AdCARE-Q, it is expected that multiple areas in the suicide prevention gatekeeper training program can be assessed and improved to deliver better quality of training and subsequently improve practices in suicide prevention. However, further studies conducting confirmatory factor analysis in a different sample would be needed to improve the psychometric qualities of the AdCARE-Q. These future studies should involve larger sample sizes, random sampling methods, different target groups and settings, exclusion of participants from the psychiatric field and participants who have received suicide

prevention training, and inclusion of better articulated items related to issues surrounding confidentiality in suicide prevention. Ideally, test–retest reliability should be evaluated as well in future studies.

**Open peer review.** To view the open peer review materials for this article, please visit http://doi.org/10.1017/gmh.2023.32.

**Supplementary material.** The supplementary material for this article can be found at https://doi.org/10.1017/gmh.2023.32.

**Data availability statement.** The data are not publicly available due to the agreement with the participants.

**Acknowledgments.** Dr. Renske Gilissen of Leiden University, Holland for permission granted to adapt the suicide prevention gatekeeper questionnaire used in the Terpstra study, 2018.

**Author contribution.** Conceptualization: L.F.C. and I.M.A.; Data analysis: I.M.A., T.I.M.D. and L.F.C.; Data collection: I.M.A., L.F.C. and A.R.O.; Methodology: L.F.C., I.M.A., K.S.P., R.R.P., Y.P.N. and T.I.M.D.; Writing – original draft preparation: I.M.A.; Writing – review and editing: L.F.C., T.I.M.D., K.S.P., R.R.P. and Y.P.N. All authors have read and agreed to the published version of the manuscript and agree to be accountable for all aspects of the work in ensuring that questions related to the accuracy or integrity of any part of the work are appropriately investigated and resolved.

**Competing interest.** The authors declare no competing interest exists.

**Ethics statement.** The study was conducted in accordance with the approved guidelines and regulations from the Research Ethics Committee of The National University of Malaysia (protocol code JEP-2020-556, 14 September 2020) and the Medical Research and Ethics Committee (MREC) Malaysia (protocol code NMRR-21-1580-60374(IIR), 22 September 2021). Informed consent was obtained from all participants involved in the study.

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
