## [Reviewer Report]

Dear Cambridge Prisms,

Thank you for the opportunity.

I am submitting the manuscript for a research article titled “The Advanced C.A.R.E. Suicide Prevention Gatekeeper Training Questionnaire For Medical Lecturers and Specialists: A Psychometric Evaluation In A Malaysian Sample”, on behalf of all the authors involved (as listed in the submission). It is a psychometric evaluation study conducted for a questionnaire that assesses knowledge gains after a suicide prevention gatekeeper training program. The target population for this study was medical lecturers and specialists, as they are the gatekeepers for medical students and the medical fraternity.

We thank you kindly for giving us the opportunity for this submission, and await the reviewers' comments for this manuscript.

Thank you.

Yours Sincerely,

Dr Iman Mohamed Ali

Psychiatrist

Department Of Psychiatry & Mental Health

Sarawak General Hospital

---

## [Reviewer Report]

1) There is a small spelling mistake in the text (line 220).

2)In line 221, the author mentioned “the sampling method described in the section above”, however, it was not stated in the manuscript. It was only mentioned at the later part of the manuscript.

3) To standardise the references.

---

## [Reviewer Report]

1. Since the questionnaire was designed to measure the efficacy of the gatekeeper training program (pre and post training), and can be used for trained and untrained gatekeepers, was any of the participants took part in any form of gatekeeper training program before the study (including AdCARE)? Could these prior trainings be a confounding factor in the outcome of this trial? If yes, suggest to elaborate under discussion or limitation.

2. Was there any exclusion criteria for sample selection?

3. What is the sampling method used? It was mentioned as purposive under the heading of strength and limitation, probably it would also be good to state in the methods for better understanding of the methodology.

4. It’s interesting to note that all the participants are having the background as medical doctors except for one from the allied health, which is the Nursing lecturer that was being categorized as “Others” department. Is the score of the allied health lecturer an outlier?

5. The questionnaire was designed to measure the efficacy of gatekeeper training program like AdCARE program, which include safety planning and suicide postvention. From the result of the trial, some of the questions pertaining to these few areas were removed, for example C9 on the specific knowledge of safety planning; while question D4 which is only a general topic talking about safety planning, was retained. What could be the impact of the removal of these questions have on the initial purpose and design of the questionnaire in measuring the efficacy of training program covering these areas? Is it still able to measure and act as a tool to improve the training program after consolidation of the questionnaire? Maybe author can discuss on these aspects.

6. “In the field of medicine, age generally signifies number of years of service or experience. The larger amount of experience with suicidal cases due to longer time spent in the field of medicine, would make medical professionals in the older age group more confident about their knowledge on suicide prevention” – is there any literature that support this view?

---

## [Reviewer Report]

Dear Editorial Team,

We have made a revision to our manuscript titled “The Advanced C.A.R.E. Suicide Prevention Gatekeeper Training Questionnaire For Medical Lecturers and Specialists: A Psychometric Evaluation In A Malaysian Sample” as per the reviewer’s comments.

We would also like to choose not to send in a graphical abstract.

We hope that this manuscript will be accepted for publication and thank you sincerely for the opportunity.

Thank You.

Yours Sincerely,

Dr. Chan Lai Fong

MD(UKM), MSc(Maastricht), MMedPsych(UKM), Clin. Fellow. Mood & Anxiety Disorders(Univ. of Toronto), 

Associate Professor and Consultant Psychiatrist,

Department of Psychiatry,

Faculty of Medicine

Hospital Canselor Tuanku Muhriz,

National University of Malaysia (UKM)

Jalan Yaacob Latif, Bandar Tun Razak,

56000 Cheras, Kuala Lumpur,

Malaysia.

Tel: +603-91456143

Fax: +603-91456681

Email: laifchan@ppukm.ukm.edu.my